# Reciprocal inhibition of YAP/TAZ and NF-κB regulates osteoarthritic cartilage degradation

Yujie Deng[1,2], Jinqiu Lu[1], Wenling Li[2], Ailing Wu[1], Xu Zhang[2], Wenxue Tong[3], Kiwai Kevin Ho[3], Ling Qin[3], Hai Song[1] & Kinglun Kingston Mak[2]

Osteoarthritis is one of the leading causes of pain and disability in the aged population due to articular cartilage damage. This warrants investigation of signaling mechanisms that could protect cartilage from degeneration and degradation. Here we show in a murine model of experimental osteoarthritis that YAP activation by transgenic overexpression or by deletion of its upstream inhibitory kinases Mst1/2 preserves articular cartilage integrity, whereas deletion of YAP in chondrocytes promotes cartilage disruption. Our work shows that YAP is both necessary and sufficient for the maintenance of cartilage homeostasis in osteoarthritis. Mechanistically, inflammatory cytokines, such as TNFα or IL-1β, trigger YAP/TAZ degradation through TAK1-mediated phosphorylation. Furthermore, YAP directly interacts with TAK1 and attenuates NF-κB signaling by inhibiting substrate accessibility of TAK1. Our study establishes a reciprocal antagonism between Hippo-YAP/TAZ and NF-κB signaling in regulating the induction of matrix-degrading enzyme expression and cartilage degradation during osteoarthritis pathogenesis.

---

[1] Life Sciences Institute and Innovation Center for Cell Signaling Network, Zhejiang University, Hangzhou 310058, China. [2] Developmental and Regenerative Biology, School of Biomedical Sciences, The Chinese University of Hong Kong, Shatin, Hong Kong SAR, China. [3] Musculoskeletal Research Laboratory, Department of Orthopaedics & Traumatology, Faculty of Medicine, The Chinese University of Hong Kong, Shatin, Hong Kong SAR, China. Correspondence and requests for materials should be addressed to H.S. (email: haisong@zju.edu.cn) or to K.K.M. (email: kingstonmak@gmail.com)

Osteoarthritis (OA) is one of the most common degenerative diseases and the incidence increases significantly with age. The disease is characterized by progressive degradation of articular cartilage, subchondral bone thickening, and osteophyte formation, which ultimately leads to loss of joint mobility and joint functions. Cartilage loss is caused by multifactorial parameters, including excessive production of matrix-degrading enzymes such as aggrecanases and matrix metalloproteinases (MMPs)[1], accelerated chondrocyte hypertrophy and increased focal calcification of joint cartilage. These conditions are commonly characterized by elevated expression of Col10a1 and alkaline phosphatase[2–4]. Eventually, cells undergo apoptosis, which leads to destruction of cartilage tissues[5]. Articular chondrocytes differ from growth plate chondrocytes as they do not normally undergo proliferation, maturation, hypertrophy, apoptosis, and ossification[6,7]. However, the molecular mechanisms regulating these processes in articular chondrocytes remain unclear. These regulatory processes are highly relevant to the onsets, pathogenesis, and progression of OA.

A variety of cytokines and chemokines are ectopically expressed in OA chondrocytes, synovial macrophages, and fibroblasts. Pro-inflammatory mediators such as tumor necrosis factor alpha (TNFα), interleukin-1 beta (IL-1β), and IL-6 are implicated in OA pathophysiology[8]. These catabolic factors activate a series of pathways including NF-κB signaling, which plays a major role in OA pathogenesis[9]. It has been shown that NF-κB signaling orchestrates mechanical, inflammatory, and oxidative stress-activated processes that contribute to cartilage tissue damage and thus representing an attractive therapeutic target for OA treatment[10–12]. A better understanding of the mechanism in modulating NF-κB signaling activity stands essential for the development of effective therapeutic intervention.

Hippo signaling is identified to control organ size and tissue regeneration in many organs[13,14]. Central to this pathway is a kinase cascade consisting of MST1/2, SAV, LATS1/2, and MOB1A/B. When the Hippo signaling is active, a series of phosphorylation events via MST and LATS kinases ultimately leads to the phosphorylation of YAP/TAZ, the key effectors of the pathway. Phosphorylated YAP is sequestered in the cytoplasm, which inhibits its transcriptional activity. By contrast, inactivation of the Hippo pathway increases YAP/TAZ nuclear translocation. Subsequently, they interact with TEADs or other transcription factors to regulate downstream signaling cascades in order to control cell proliferation, apoptosis, differentiation, and maturation[15]. We have shown that Hippo pathway mediates its effect through YAP in regulating chondrocyte differentiation at multiple steps during endochondral ossification and bone repair. YAP promotes chondrocyte proliferation but inhibits subsequent maturation by binding with different transcription factors implicated in chondrocyte differentiation[16]. Whether Hippo pathway or YAP regulates articular cartilage homeostasis similar to that of skeletal development remains elusive. Previous studies have firmly established pivotal role of Hippo-YAP/TAZ pathway in embryonic development, tissue homeostasis, and tumorigenesis. Recently, several studies have uncovered novel roles of Hippo signaling in regulating innate immunity, autoimmunity, and cancer immunity[17–19]. However, whether the Hippo-YAP/TAZ pathway plays a role in regulating inflammatory response during OA pathogenesis remains elusive.

Here, we investigated the roles of Hippo pathway and YAP in maintaining articular cartilage integrity during OA pathogenesis. We found that Hippo signaling mediates its signals through YAP to control articular cartilage homeostasis. YAP is necessary and sufficient to attenuate OA progression by inhibiting inflammatory responses triggered by NF-κB signaling. Furthermore, inflammatory cytokines activates Hippo signaling and promotes YAP

phosphorylation mediated by TAK1 and association with β-TRCP for proteasome-mediated degradation. Our findings suggest that targeting YAP is a viable strategy for treating OA.

## Results

**Reduced expression of YAP in osteoarthritic cartilage.** To investigate the function of YAP in articular cartilage maintenance, we first examined the endogenous expression of YAP, a key mediator of Hippo signaling, in the knee joints from 1- to 6-month-old wild-type mice (Fig. 1a, b). When the mice were young at 1- and 2-month-old, strong YAP expression was observed in all zones of the articular cartilage. As the mice aged, we found that YAP expression was gradually reduced and its expression was remarkably lower in the 6-month-old mice. These data suggest a gradual reduction of YAP in articular chondrocyte maturation. As OA is one of the ageing related diseases, we thus recapitulated the degenerative condition of articular cartilage by performing surgically induced OA in adult wild-type mice. We found that injured articular cartilage also showed significant reduction of YAP expression and its expression level correlated coherently to the severity of cartilage degradation in OA conditions as shown by knee joint sections at 4 and 10 weeks post operation respectively (Fig. 1c, d). Furthermore, YAP expression levels were concomitantly reduced according to the severity and OARSI grade of OA in human patient samples[20] (Fig. 1e, f). Altogether, our findings suggest that YAP expression is highly correlated to the pathogenesis of OA.

**YAP attenuates cartilage degradation during OA progression.** To elucidate the functional role of YAP in OA pathogenesis in vivo, we first generated $Mst1^{f/f};Mst2^{f/f};Col2a1$-Cre mutant mice in which Hippo signaling is inactivated in chondrocytes. The mutant mice were phenotypically normal with no obvious skeletal defect including articular cartilage, albeit YAP expression was strongly upregulated in articular cartilage (Supplementary Figure 1a, b). Next, we surgically induced osteoarthritic condition in the mutant mice and examined the articular cartilage degradation. Two months after surgery, we found that the integrity of articular cartilage of the $Mst1^{f/f};Mst2^{f/f};Col2a1$-Cre mutant mice was maintained significantly better than that of the control group under both ACLT (Anterior Cruciate Ligament Transection) and DMM (Destabilization of the Medial Meniscus) surgical conditions (Fig. 2a, b and Supplementary Figure 1c). The expression of YAP was maintained at a higher level in the mutant mice, but Mmp13 (matrix metalloproteinases 13) expression was significantly lower (Fig. 2c, d). In addition, inflammation was less severe in the peripheral fibrous tissues with fewer synovial lining cells around the knee joint of the mutant mice as shown by F4/80 and CD11b expression (Supplementary Figure 1d). Next, we isolated primary chondrocytes from the newborn mutant mice and treated them with TNFα in culture. Consistent with the in vivo phenotypes, the ECM (extracellular matrix) degradation enzymes, such as Mmps and Adamts4/5 (a disintegrin and metalloproteinase with thrombospondin motifs 4/5), were significantly upregulated, while the expression of the cartilage extracellular matrix components (Col2a1 and Aggrecan), Yap1/Taz and their target genes was greatly inhibited in response to TNFα treatment in control chondrocytes (Fig. 2e and Supplementary Figure 1e). However, deletion of Mst1/2 in chondrocytes attenuated these effects induced by TNFα (Fig. 2e and Supplementary Figure 1e). Collectively, our data indicate that activation of YAP in articular chondrocytes attenuates OA progression.

To further determine the functions of YAP in protecting cartilage degradation during OA progression, we overexpressed YAP in

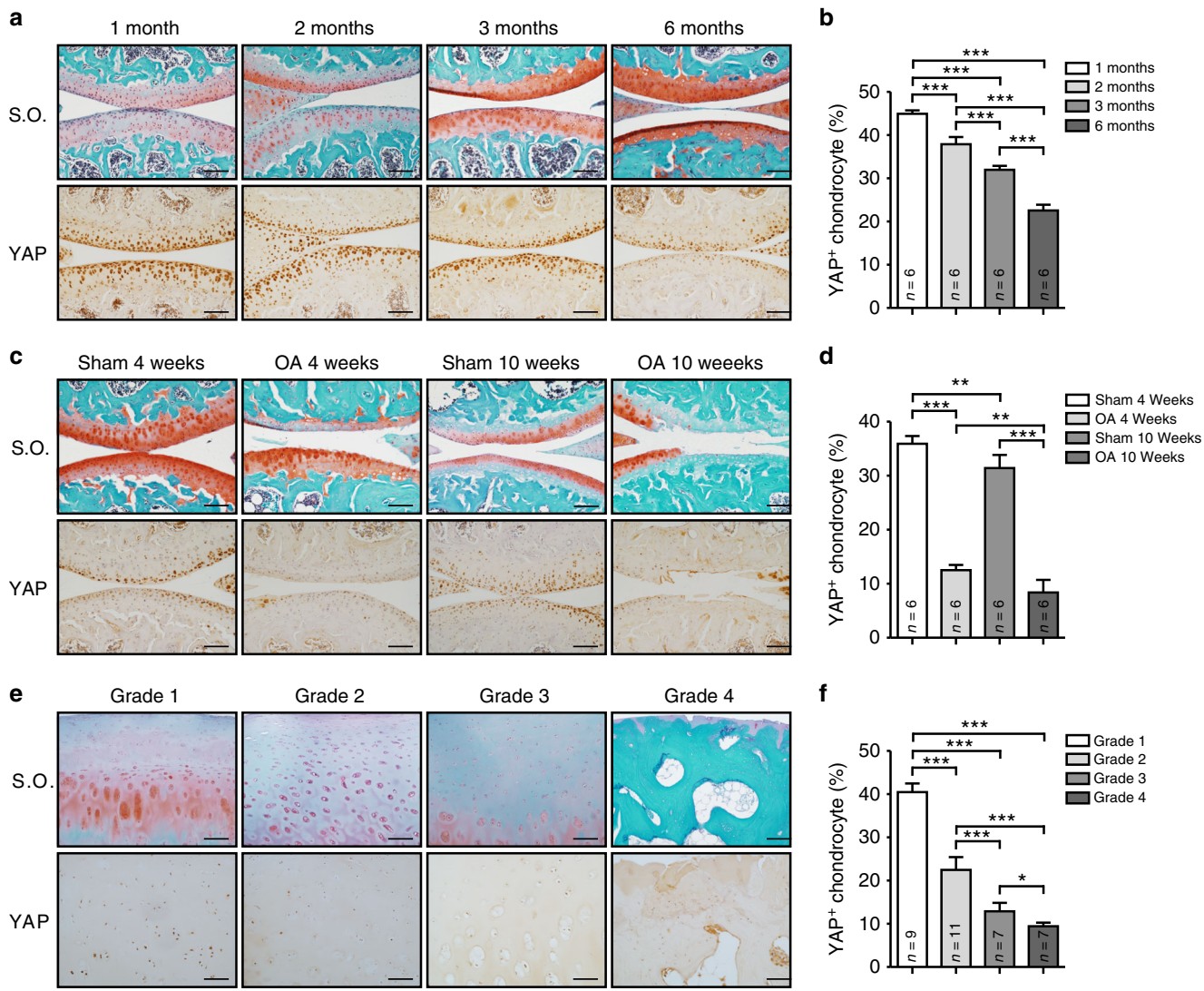

**Fig. 1** YAP expression is decreased in postnatal cartilage growth and osteoarthritic cartilages. **a** Safranin O staining (Top) and immunohistochemistry analysis of YAP expression (Bottom) in wild-type mice with age as indicated. Scale bars, 100 μm. **b** Statistical analysis of the percentage of YAP+ chondrocytes (brown) in articular cartilage from (**a**) ($n = 6$ per group). **c** Safranin O staining (Top) and immunohistochemistry analysis of YAP expression (Bottom) of 10-week-old wild-type mice after 4 or 10 weeks anterior cruciate ligament transection (ACLT) surgery respectively. Scale bars, 100 μm. **d** Statistical analysis of the percentage of YAP+ chondrocytes (brown) in articular cartilage from (**c**) ($n = 6$ per group). **e** Safranin O staining (Top) and immunohistochemistry analysis of YAP expression (Bottom) in articular cartilage of human patients with different grade of OA. Scale bars, 100 μm. **f** Statistical analysis of the percentage of YAP chondrocytes (brown) in articular cartilages from **e**. Numbers of samples examined are indicated at the bottom of the graphs. All data are presented as mean ± SD. *$p < 0.05$, **$p < 0.01$, ***$p < 0.001$, with One-way ANOVA followed by Tukey's test

chondrocytes by generating $Col2a1\text{-}Yap1^{tg/+}$ transgenic mice[16]. No obvious skeletal phenotype was observed in the articular cartilage of Yap1 heterozygous transgenic mice (Supplementary Figure 2a, b). Under osteoarthritic condition, $Col2a1\text{-}Yap1^{tg/+}$ transgenic mice displayed remarkably better cartilage integrity than that of the wild-type mice (Fig. 2f, g and Supplementary Figure 2c). In addition, Mmp13 expression in the articular chondrocytes of $Col2a1\text{-}Yap1^{tg/+}$ transgenic mice was significantly lower and less inflammatory cells were recruited as shown by F4/80 and CD11b expression in the synovial lining cells of $Col2a1\text{-}Yap1^{tg/+}$ transgenic mice (Fig. 2h, i and Supplementary Figure 2d). TNFα or IL1β treatment in transgenic chondrocytes led to lower induction of matrix-degrading enzymes, but higher expression of cartilage ECM and YAP target genes than those of the wild-type chondrocytes (Fig. 2j and Supplementary Figure 2e, f). Our results suggest that YAP protects articular cartilage from degradation during OA pathogenesis.

**Loss of YAP exaggerates cartilage destruction during OA.** To examine whether YAP is required to protect articular cartilage integrity during OA pathogenesis, we genetically removed $Yap1$ in chondrocytes by generating $Yap1^{f/f};Col2a1\text{-}Cre$ mutant mice. No obvious cartilage defect in the articular cartilage was observed in the adult $Yap1^{f/f};Col2a1\text{-}Cre$ mutant mice (Supplementary Figure 3a, b). However, under OA condition, cartilage degradation was significantly more severe than that of the control mice (Fig. 3a, b). Immunohistochemical staining revealed substantial reduction of YAP expression and elevated Mmp13 expression in the articular cartilage of the $Yap1^{f/f};Col2a1\text{-}Cre$ mutant mice under osteoarthritic condition as compared to the control animals (Fig. 3c, d). Accordingly, TNFα treatment led to increased expression of matrix-degrading enzymes and decreased expression of ECM components and YAP target genes in the primary chondrocytes from $Yap1^{f/f};Col2a1\text{-}Cre$ newborn mice (Fig. 3e, f). In addition, inflammatory cells were accumulated in the synovium of $Yap1^{f/f};Col2a1\text{-}Cre$ mice

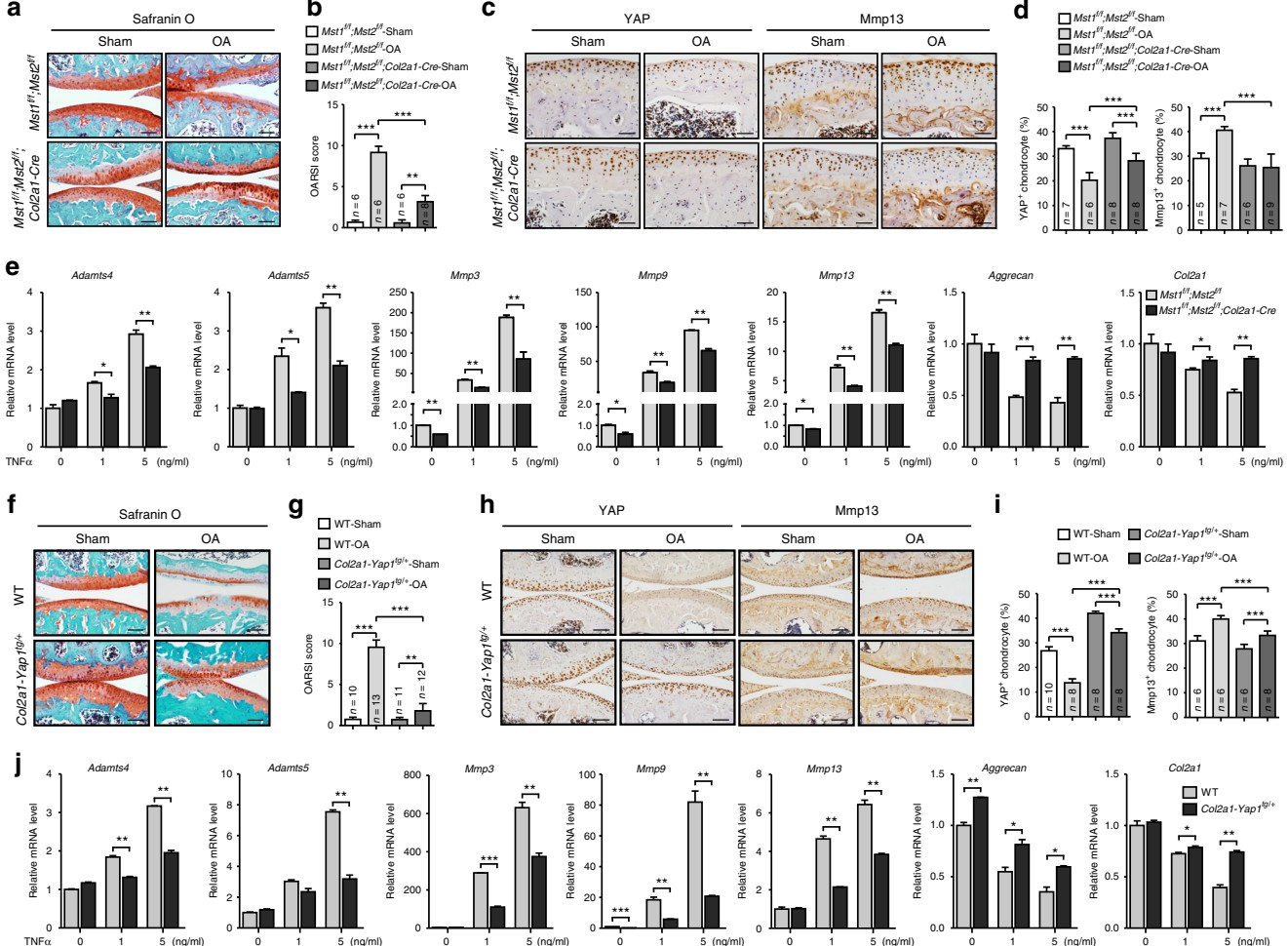

**Fig. 2** Activation of YAP attenuates cartilage degradation during OA progression. **a** Safranin O staining of sagittal sections of knee joints of *Mst1f/f; Mst2f/f; Col2a1-Cre* mice 2 months after ACLT surgery. Scale bars, 100 μm. **b** OARSI scores of samples shown in **a**. **c** Immunohistochemistry of YAP and Mmp13 expression (brown) in the knee joints of the articular cartilage of mice 2 months after ACLT surgery with genotypes as shown. Scale bars, 50 μm. **d** Statistical analysis of the percentage of YAP⁺ or Mmp13⁺ chondrocytes in articular cartilage of samples shown in **c**. **e** Gene expression analysis of matrix-degrading enzymes (*Adamts4*, *Adamts5*, *Mmp3*, *Mmp9*, and *Mmp13*) and extracellular matrix components (*Aggrecan* and *Col2a1*) in primary articular chondrocytes treated with TNFα for 24 h. **f** Safranin O staining of sagittal sections of the knee joints of *Col2a1-Yap1tg/+* transgenic mice 2 months after ACLT surgery. Scale bars, 100 μm. **g** OARSI scores of samples shown in **f**. **h** Immunohistochemistry of YAP or Mmp13 expression in articular cartilage of mice 2 months after ACLT surgery with genotypes as shown. Scale bars, 100 μm. **i** Statistical analysis of the percentage of YAP⁺ or Mmp13⁺ chondrocytes in articular cartilage of samples shown in **h**. **j** Gene expression analysis of matrix-degrading enzymes and extracellular matrix components in articular chondrocytes from WT and *Col2a1-Yap1tg/+* transgenic mice treated with TNFα for 24 h. Numbers of samples examined are indicated at the bottom of the bar charts. All data are presented as mean ± SD. *$p < 0.05$, **$p < 0.01$, ***$p < 0.001$. **b**, **d**, **g**, **i** One-way ANOVA followed by Tukey's test was performed. **e**, **j** Two-tailed Student's *t*-test was performed with experiments repeated three times independently

with OA (Fig. 3g), which are mirror results as compared to that of the *Col2a1-Yap1tg/+* transgenic mice. Altogether, our results indicate that YAP is necessary and sufficient to protect cartilage degradation in OA condition.

Next, we asked whether pharmacological treatment with agents that regulate YAP activity affects OA progression similarly to the genetic manipulation of the Hippo pathway. Lysophosphatidic acid (LPA) has been shown to inhibit LATS1/2 thereby activating YAP[21]. Conversely, verteporfin (VP) binds to YAP and inhibits its interaction with TEADs[22]. We first validated the effects of these agents in primary chondrocytes. As expected, LPA treatment inhibited YAP phosphorylation and stabilized YAP proteins in primary chondrocytes (Supplementary Figure 4a), whereas VP treatment inhibited YAP activities and thereby led to its degradation (Supplementary Figure 4b). Next, we examined their effects in articular cartilage in vivo under OA condition.

Consistent with our mouse genetic models, alginate beads soaked with either LPA or VP and implanted into the articular joint cavity of wild-type mice immediately after OA surgery exhibited similar effects on articular cartilage integrity as *Yap1* transgenic or knockout mice, respectively (Supplementary Figure 4c–f). Moreover, co-treatment of IL-1β or TNFα with LPA or VP in culture showed consistent results in the expression of matrix-degrading enzymes and ECM, respectively, as the *Mst1/2* or *Yap1* mutant chondrocytes (Supplementary Figure 4g, h). Thus, our data indicate that pharmacological activation of YAP protects osteoarthritic cartilage degradation.

**Inflammatory cytokines promote YAP degradation.** As proinflammatory cytokines, such as TNFα, IL-1β, and IL-6, are implicated in OA pathophysiology[9,23], the effect of these inflammatory cytokines on Hippo signaling was examined in

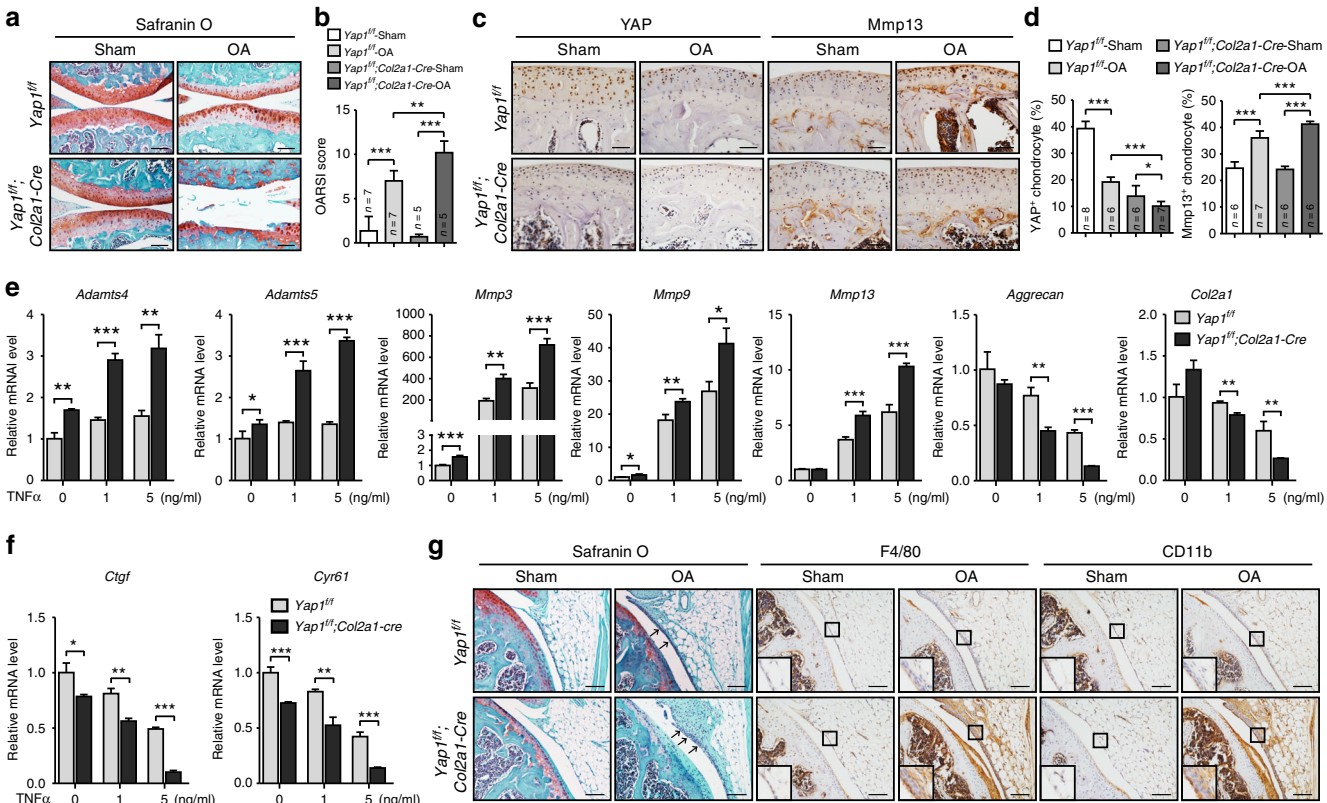

**Fig. 3** Loss of YAP exaggerates cartilage destruction during OA. **a** Safranin O staining of sagittal sections of the knee joints of *Yap1f/f; Col2a1-Cre* mice 2 months after ACLT surgery. Scale bars, 100 μm. **b** OARSI scores of samples shown in **a**. **c** Immunohistochemistry of YAP or Mmp13 expression in articular cartilage of mice 2 months after ACLT surgery with genotypes as shown. Scale bars, 50 μm. **d** Statistical analysis of the percentage of YAP+ or Mmp13+ chondrocytes in articular cartilage of samples shown in **c**. **e** Gene expression analysis of matrix-degrading enzymes and extracellular matrix components in articular chondrocytes treated with TNFα for 24 h. **f** Gene expression analysis of YAP target genes in primary chondrocytes after treatment with TNFα for 24 h. **g** Safranin O staining (Left) and immunohistochemistry analysis of macrophage marker genes F4/80 (Middle) or CD11b (Right) of the synovial membrane of mice with genotypes as shown after 2 months after OA surgery. Inflammation was more severe in the peripheral fibrous tissues with hyperplasia synovial lining cells (black arrows) in the synovial joint of the *Yap1f/f; Col2a1-Cre* mice with OA. Scale bars, 100 μm (*n* = 3 per group). Numbers of mice examined are indicated at the bottom of the graphs. All data are presented as mean ± SD. *$p < 0.05$, **$p < 0.01$, ***$p < 0.001$. **b, d** One-way ANOVA followed by Tukey's test was performed. **e, f** Two-tailed Student's *t*-test was performed with experiments repeated three times independently

primary chondrocytes to mimic the pathological conditions of OA. We found that the expression of *Mst1/2* and *Lats1/2* displayed no obvious changes upon treatment with TNFα or IL-1β (Supplementary Figure 5a). Notably, the expression of *Yap1/Taz* and their target genes *Ctgf* and *Cyr61* was reduced in response to TNFα treatment in primary chondrocytes (Fig. 4a and Supplementary Figure 5a). Consistently, Gal4/TEAD4-luciferase reporter activity was inhibited in response to TNFα treatment in a dose dependent manner in primary chondrocytes (Fig. 4b). Moreover, TNFα treatment also suppressed YAP-induced Gal4/TEAD4-luciferase reporter activity in HEK293A cells (Fig. 4c). However, at protein level, MST1 and LATS1 were strongly activated as shown by increased phosphorylation levels shortly after TNFα or IL-1β stimulation, respectively (Fig. 4d). The phosphorylation levels of both YAP and TAZ were greatly increased (Fig. 4d). Moreover, we found that the total protein levels of YAP and TAZ were reduced after TNFα or IL-1β treatment in primary chondrocytes (Fig. 4e). As ubiquitin-mediated proteolysis is one of the most common mechanisms for protein degradation, we examined whether YAP is removed through proteasome upon TNFα stimulation. We found that YAP protein levels were restored upon proteasome inhibitor MG132 treatment, but not lysosomal inhibitor chloroquine (Chlq) (Fig. 4f). In addition, TNFα treatment resulted in remarkable poly-ubiquitination

modification and degradation of YAP (Fig. 4g, h). Furthermore, immunofluorescence analysis revealed that YAP was exported to the cytoplasm after TNFα stimulation in primary chondrocytes or in IL-1β-treated HeLa cells (Fig. 4i and Supplementary Figure 5b). Taken together, our data indicate that inflammatory cytokines trigger Hippo pathway activation and promote proteasomal degradation of YAP.

**TAK1 associates with and phosphorylates YAP/TAZ independent of LATS.** The direct phosphorylation of YAP/TAZ by LATS1/2 kinases leads to YAP/TAZ cytoplasmic retention and degradation mediated by SCF/β-TRCP E3 ubiquitin ligase in response to various stimuli[24,25]. We asked whether TNFα-induced YAP/TAZ phosphorylation and degradation are dependent on LATS1/2. Intriguingly, a decrease of YAP/TAZ protein levels was still observed in LATS1/2 DKO (double knockout) HEK293A cells in response to TNFα stimulation (Fig. 5a). In addition, the reduced YAP/TAZ protein levels were restored when cells were treated with MG132, but not chloroquine (Fig. 5a and Supplementary Figure 6a). These results suggest that TNFα-induced proteasomal degradation of YAP/TAZ is independent of LATS1/2 kinases. As we observed an increased phosphorylation of YAP/TAZ upon TNFα treatment in chondrocytes, this prompted us to test whether kinases in the TNFα-activated

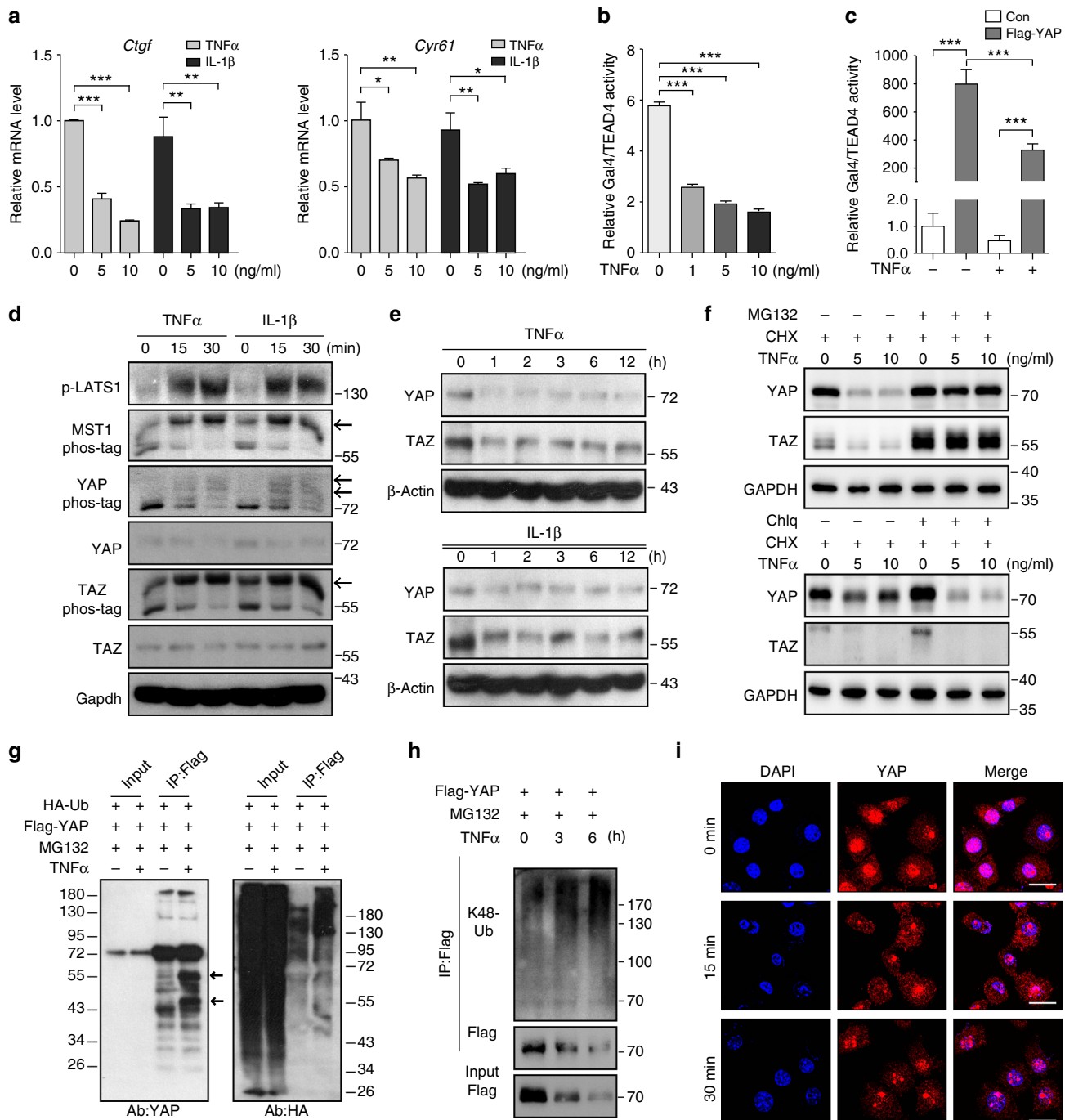

**Fig. 4** Inflammatory cytokines trigger Hippo signaling activation and YAP degradation. **a** Gene expression analysis of *Ctgf* and *Cyr61* in primary chondrocytes after treatment with TNFα or IL-1β for 24 h, respectively. **b** Luciferase reporter assay of Gal4/TEAD4-reporter in primary chondrocytes treated with different doses of TNFα for 24 h. **c** Luciferase analysis of Gal4/TEAD4-reporter activity in HEK293A cells transfected with Flag-tagged YAP for 24 h and subjected to TNFα treatment for 6 h. **d** Phos-tag gel and immunoblot analyses of Hippo components in primary articular chondrocytes isolated from newborn mice after treatment with TNFα or IL-1β at 5 ng/ml, respectively. Arrows indicate the phosphorylated form of proteins. **e** Western blot analysis of YAP and TAZ in primary articular chondrocytes treated with TNFα or IL-1β at 5 ng/ml respectively for the indicated time. **f** Western blot analysis of YAP and TAZ expression in HEK293A cells with pre-treatment of CHX for 2 h followed by treatment with TNFα for 6 h together with MG132 or Chlq (chloroquine). **g** Primary chondrocytes were transfected with indicated plasmids and harvested for ubiquitination analysis 6 h after treatment with MG132 at 10 μM and TNFα at 5 ng/ml. Arrows indicate the degraded YAP. **h** Ubiquitination analysis of YAP after TNFα treatment at 5 ng/ml for the indicated time in HEK293T cells. **i** Immunofluorescence staining of YAP in primary chondrocytes after treatment with TNFα at 5 ng/ml. Scale bars, 20 μm. All data are presented as mean ± SD. *$p < 0.05$, **$p < 0.01$, ***$p < 0.001$. **a**, **b** One-way ANOVA followed by Dunnett's test was performed with 0 ng/ml group as control. **c** One-way ANOVA followed by Tukey's test was used. All experiments were repeated three times independently

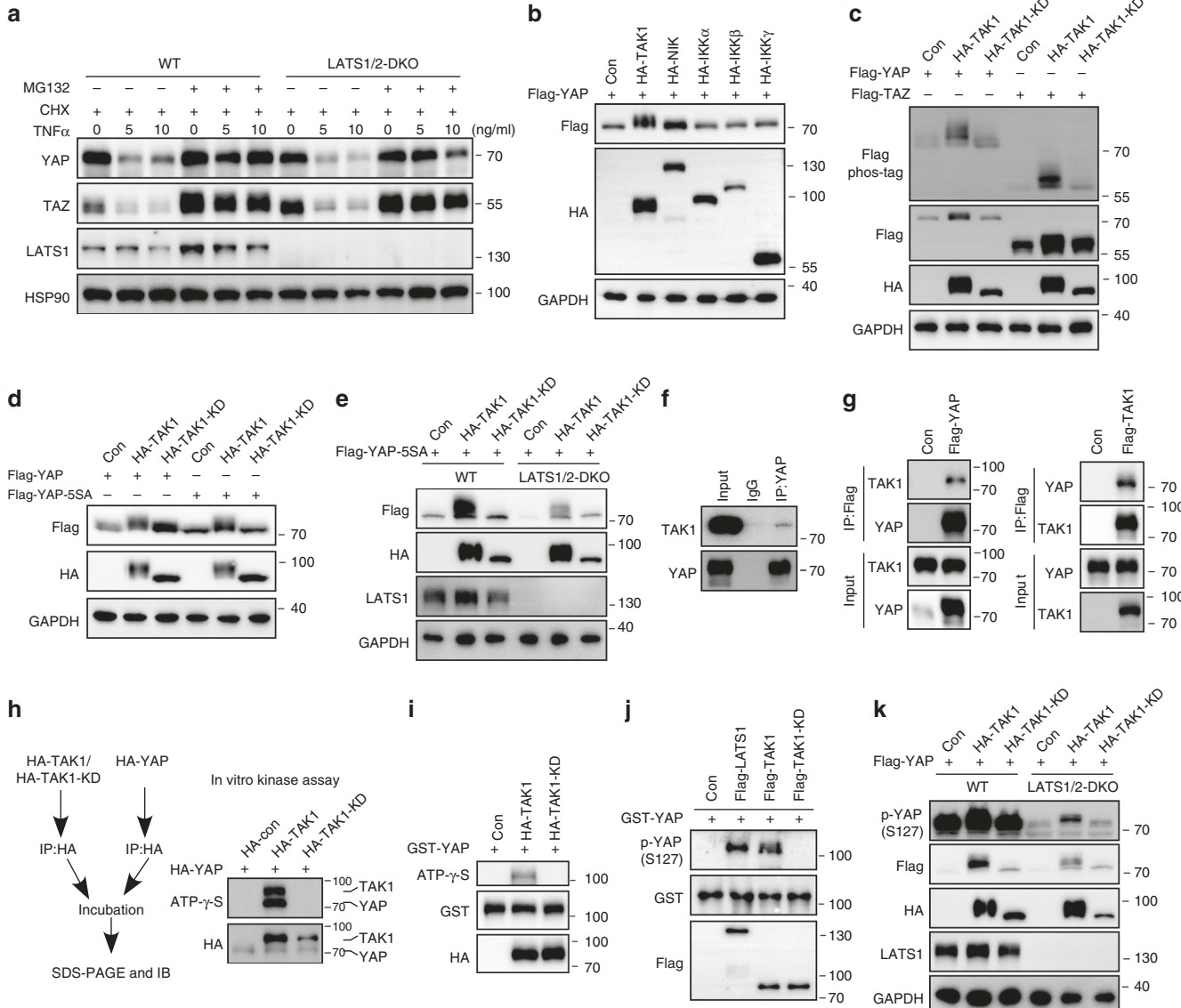

**Fig. 5** TAK1 associates with and phosphorylates YAP/TAZ independent of LATS1/2. **a** Western blot analysis of YAP and TAZ in WT and LATS1/2 DKO HEK293A cells treated with TNFα and MG132 as indicated. **b** Western blot analysis of lysate of HEK293A cells transfected with Flag-tagged YAP and indicated kinases. **c** Phos-tag SDS-PAGE analysis of lysate of HEK293T cells transfected with Flag-tagged YAP and indicated kinases. **d** Western blot analysis of lysate of HEK293T cells transfected with Flag-tagged YAP or YAP5SA and indicated kinases. **e** Western blot analysis of lysate of WT and LATS1/2 DKO HEK293A cells transfected with Flag-tagged YAP-5SA mutant with indicated kinases. **f** Immunoprecipitation assay using anti-YAP antibody to detect endogenous complex of YAP and TAK1 in HEK293T cells. **g** Immunoprecipitation assay of Flag-tagged TAK1 or YAP protein expressed in HEK293T cells. **h, i** Wild-type, TAK1-KD and YAP purified from HEK293T cells (**h**) or recombinant GST-YAP from *E.coli* (**i**) were used for in vitro kinase assay. **j** Purified LATS1, wild-type and kinase-dead TAK1 from HEK293T cells, and recombinant GST-YAP were used for in vitro kinase assay. **k** Western blot analysis of cell lysate of wild-type and LATS1/2 DKO HEK293A cells transfected with wild-type or TAK1-KD and Flag-tagged YAP plasmids. All experiments were repeated three times independently

NF-κB pathway are involved in the phosphorylation of YAP/TAZ. We co-expressed YAP with a series of kinases related to the NF-κB pathway in HEK293T cells (Fig. 5b). Of note, Flag-tagged YAP or TAZ displayed a significant mobility shift only when TAK1 kinase was co-expressed (Fig. 5b and Supplementary Figure 6b). Moreover, the kinase-dead mutant of TAK1 (TAK1-KD) was unable to promote the mobility shift of YAP or TAZ in phos-tag gel assay (Fig. 5c). These results suggest that YAP/TAZ are the targets of TAK1 for phosphorylation modification. In addition, overexpression of YAP-5SA where all LATS1/2 kinases phosphorylation sites were mutated, also displayed a remarkable mobility shift on SDS-PAGE in the presence of TAK1 kinase in wild-type or LATS1/2 DKO cells respectively (Fig. 5d, e).

These data indicate that TAK1 phosphorylates YAP through a LATS1/2 kinases independent mechanism.

To further investigate the molecular mechanisms how TAK1 regulates YAP activity, we tested whether TAK1 directly associates with and phosphorylates YAP. By co-immunoprecipitation assay, we observed an interaction of TAK1 and YAP in endogenous proteins or using overexpression of reciprocal tagged proteins in HEK293T cells (Fig. 5f, g and Supplementary Figure 6c). Next, we used an in vitro kinase assay to test whether TAK1 directly phosphorylates YAP. We found that TAK1 purified from HEK293T cells phosphorylated YAP purified from a separate set of HEK293T cells or GST-YAP recombinant proteins isolated from *Escherichia coli* (Fig. 5h, i).

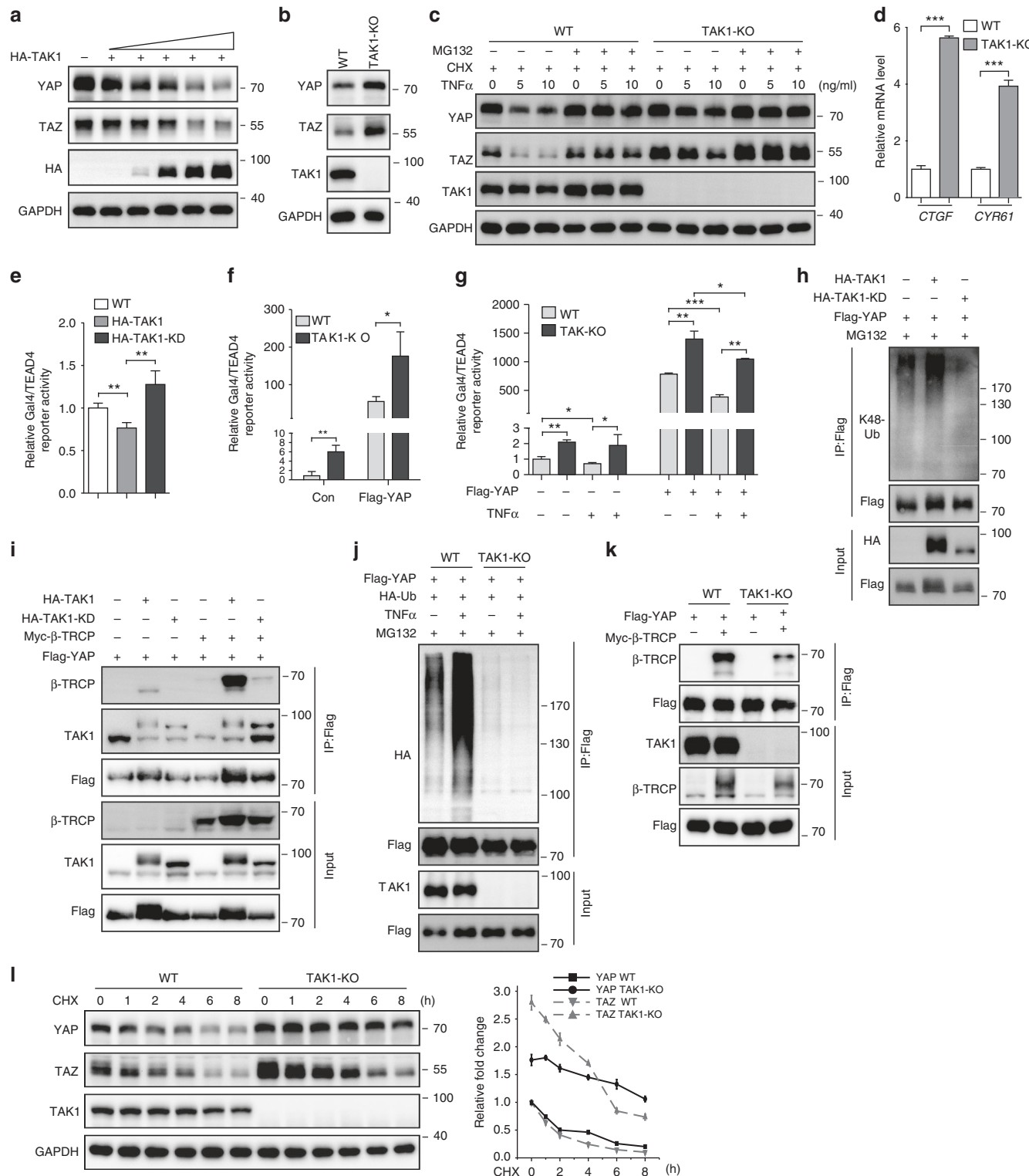

Since phosphorylation of YAP at S127 leads to its cytoplasmic retention, and we found that TNFα treatment stimulates YAP translocation into cytoplasm (Fig. 4i and Supplementary Figure 5b). These observations prompted us to test whether TAK1 phosphorylates YAP at S127. Our in vitro kinase assay showed that S127 of GST-YAP recombinant protein was phosphorylated in the presence of TAK1, but not TAK1-KD (Fig. 5j). Furthermore, expression of TAK1 enhanced S127 phosphorylation of Flag-tagged YAP in LATS1/2 DKO HEK293A

cells (Fig. 5k). To identify additional and potential phosphorylation sites on YAP by TAK1, we next performed mass spectrometry analysis of Flag-tagged YAP co-expressed with HA-tagged TAK1 or TAK1-KD. We observed a substantial increased percentage of several phosphorylation sites of YAP in the presence of TAK1 (Supplementary Figure 6d and Supplementary Dataset 1, 2). We subsequently generated a series of YAP mutations targeting these sites (Supplementary Figure 6e). However, we did not identify specific sites targeted by TAK1 by

**Fig. 6** TAK1 inhibits YAP activity through β-TRCP-mediated ubiquitination and proteasomal degradation. **a** Immunoblot analysis of endogenous YAP in HEK293T cells transfected with increasing TAK1 plasmid. **b** Western blot analysis of lysate of wild-type and TAK1-KO HEK293T cells. **c** Western blot analysis of lysate of wild-type and TAK1-KO HEK293A cells treated with MG132, CHX, and TNFα as indicated. **d** Gene expression analysis of YAP target genes in wild-type or TAK1-KO HEK293T cells by qRT-PCR assay. **e** Luciferase assay of Gal4/TEAD4 reporter activity after transfection of wild-type or TAK1-KD plasmid in HEK293T cells. **f** Luciferase assay of Gal4/TEAD4 reporter activity after transfection of control or YAP plasmid in wild-type or TAK1-KO HEK293T cells. **g** Luciferase assay of Gal4/TEAD4 reporter activity after treated with TNFα for 6 h in wild-type or TAK1-KO HEK293A cells transfected with control or YAP plasmid. **h** Ubiquitination assay of Flag-tagged YAP with overexpression of TAK1 or TAK1-KD in HEK293T cells detected by anti-K48-Ubiquitin antibody. **i** Immunoprecipitation assay of YAP and β-TRCP interaction with overexpression of TAK1 or TAK1-KD in HEK293T cells. **j** Ubiquitination assay of Flag-tagged YAP in wild-type or TAK1-KO HEK293A cells with overexpression of HA-tagged Ubiquitin after treatment with MG132 (10 μM) for 2 h and then treated with TNFα at 5 ng/ml for 4 h. **k** Immunoprecipitation assay of YAP and β-TRCP interaction in wild-type or TAK1-KO HEK293A cells. **l** Analysis of YAP/TAZ stability in wild-type and TAK1-KO HEK293T cells by CHX chase experiments. Protein synthesis was blocked by treatment with 50 μg/ml CHX for the indicated time. Quantification of the relative expression level of YAP/TAZ related to GAPDH in samples are shown in the right panel. All experiments were repeated three times independently and all data are presented as mean ± SD. *$p < 0.05$, **$p < 0.01$, ***$p < 0.001$. **e**, **g** One-way ANOVA followed by Tukey's test was performed. **d**, **f** Two-tailed Student's $t$-test was performed

examining the mobility shift of YAP induced by TAK1. Thus, our data indicate that TAK1 directly phosphorylates YAP at multiple sites.

**TAK1 inhibits YAP activity through β-TRCP.** Next, we tested whether TAK1 promotes YAP degradation. We found that TAK1 expression triggered degradation of endogenous YAP and TAZ in a dose dependent manner in HEK293T cells (Fig. 6a). Conversely, the basal levels of YAP and TAZ were increased in TAK1-KO HEK293T cells (Fig. 6b). More importantly, TNFα-induced degradation of YAP/TAZ was significantly compromised in the TAK1-KO HEK293A cells (Fig. 6c). Next, we investigated whether TAK1 modulates YAP/TEAD transcriptional activity. As expected, there was an increased expression of YAP target genes, such as *CTGF* and *CYR61*, in TAK1-KO HEK293T cells (Fig. 6d). Expression of TAK1 inhibited Gal4/TEAD4-luciferase reporter activity as compared to the control and TAK1-KD mutant (Fig. 6e). Furthermore, YAP induced a higher reporter activity in TAK1-KO HEK293T cells (Fig. 6f). Consistently, TNFα-induced reduction of reporter activity was also compromised in TAK1-KO HEK293T cells (Fig. 6g). These results indicate that TAK1 promotes YAP degradation and suppresses its activity.

To further explore how TAK1 contributes to YAP degradation, we first examined whether TAK1 promotes YAP ubiquitination. We found that the expression of TAK1 enhanced K48-linked poly-ubiquitination modification of YAP protein (Fig. 6h). Previous study demonstrates that Hippo-regulated YAP degradation is through SCF-β-TRCP-mediated ubiquitination[25]. Therefore, we tested whether β-TRCP is involved in YAP degradation regulated by TAK1. Notably, expression of TAK1, but not TAK1-KD mutant, greatly enhanced the interaction of YAP with β-TRCP (Fig. 6i). Conversely, knockout of TAK1 inhibited YAP poly-ubiquitination modification and abrogated YAP–β-TRCP interaction (Fig. 6j, k). In addition, endogenous YAP and TAZ proteins were remarkably stabilized in the TAK1-KO HEK293T cells, but degraded quickly over time in HEK293T cells in a CHX (Cycloheximide) chase experiment (Fig. 6l). Together, our results support the functional importance of TAK1 in inhibition of YAP activity by stimulating β-TRCP-mediated YAP ubiquitination and degradation during inflammatory cytokine stimulation.

**YAP attenuates NF-κB pathway by inhibiting IKKα/β activation.** One of the complications for OA pathogenesis is inflammation[23]. Inflammatory cytokines TNFα and IL-1β are implicated in cartilage degradation through regulating NF-κB and JNK signaling pathways[11,12,26]. We asked whether YAP modulates the activity of TNFα signaling and its subsequent actions in regulating articular cartilage integrity during OA condition. In

primary chondrocytes, overexpression of YAP inhibited TNFα-induced JNK and p65 activation (Supplementary Figure 7a). Treatment of JNK inhibitor SP600125 only inhibited TNFα-induced upregulation of Adamts5 and Mmp3. However, treatment of TAK1 inhibitor 5Z-7-Oxozeaenol (5Z-7-O) greatly inhibited all TNFα or IL1β effects and evoked a broad suppression of Adamts and Mmp expression (Supplementary Figure 7b, c and d). In *Col2a1-Yap1tg/+* transgenic chondrocytes, we found that NF-κB signaling activity was greatly attenuated after TNFα stimulation as shown by reduced phosphorylation of p65 while JNK signaling was only slightly inhibited by YAP (Fig. 7a and Supplementary Figure 7e). To further verify that YAP modulates NF-κB signaling activity, we examined the cellular localization of p65 in response to YAP overexpression. p65 was translocated into the nucleus under IL-1β treatment in primary chondrocytes (Fig. 7b). However, when YAP was overexpressed, IL-1β-induced nuclear translocation of p65 was significantly inhibited (Fig. 7b). More importantly, a significantly reduced expression of phosphorylated p65 in the *Col2a1-Yap1tg/+* mutant chondrocytes under osteoarthritic condition was observed as compared to the controls (Fig. 7c). These data strongly suggest that YAP expression attenuates NF-κB signaling to protect cartilage degradation during OA.

To explore the molecular basis of YAP/TAZ in regulating the activity of NF-κB pathway, we first examined the effect of YAP/TAZ on NF-κB luciferase reporter. As measured by the NF-κB luciferase reporter, we observed a profound inhibition of TNFα-induced NF-κB transactivation by both YAP-5SA and transcriptionally inactive YAP-6SA, similar to that of wild-type YAP (Fig. 7d). Furthermore, YAP greatly suppressed TAK1-, IKKα/β- (inhibitor of nuclear factor kappa-B kinase subunit alpha/beta), or IKKγ-induced NF-κB reporter activity (Fig. 7e–h,). However, YAP only slightly inhibited p65-induced NF-κB reporter activity (Fig. 7i,). These data suggest that YAP acts upon the components upstream of p65. Similarly, TAZ exhibited consistent inhibitory effects as YAP on NF-κB luciferase reporter assays (Supplementary Figure 8a-e). These observations suggest that YAP/TAZ-mediated suppression on NF-κB activity might be a direct effect rather than through their transcriptional targets.

To further explore the mechanism of how YAP inhibits NF-κB signaling activity, we examined whether YAP interacts with the components upstream of p65 in the NF-κB pathway. Intriguingly, an interaction of YAP with IKKα/β and IKKγ was detected by co-immunoprecipitation assays (Fig. 7j). Next, we examined whether YAP inhibits NF-κB activity through suppressing the TAK1-IKKs cascade activation. TRAFs-mediated TAK1 K63-linked poly-ubiquitylation is critical for TAK1 activation. We found that K63-linked poly-ubiquitylation and phosphorylation of TAK1 were not affected by YAP overexpression (Supplementary Fig. 8f, g).

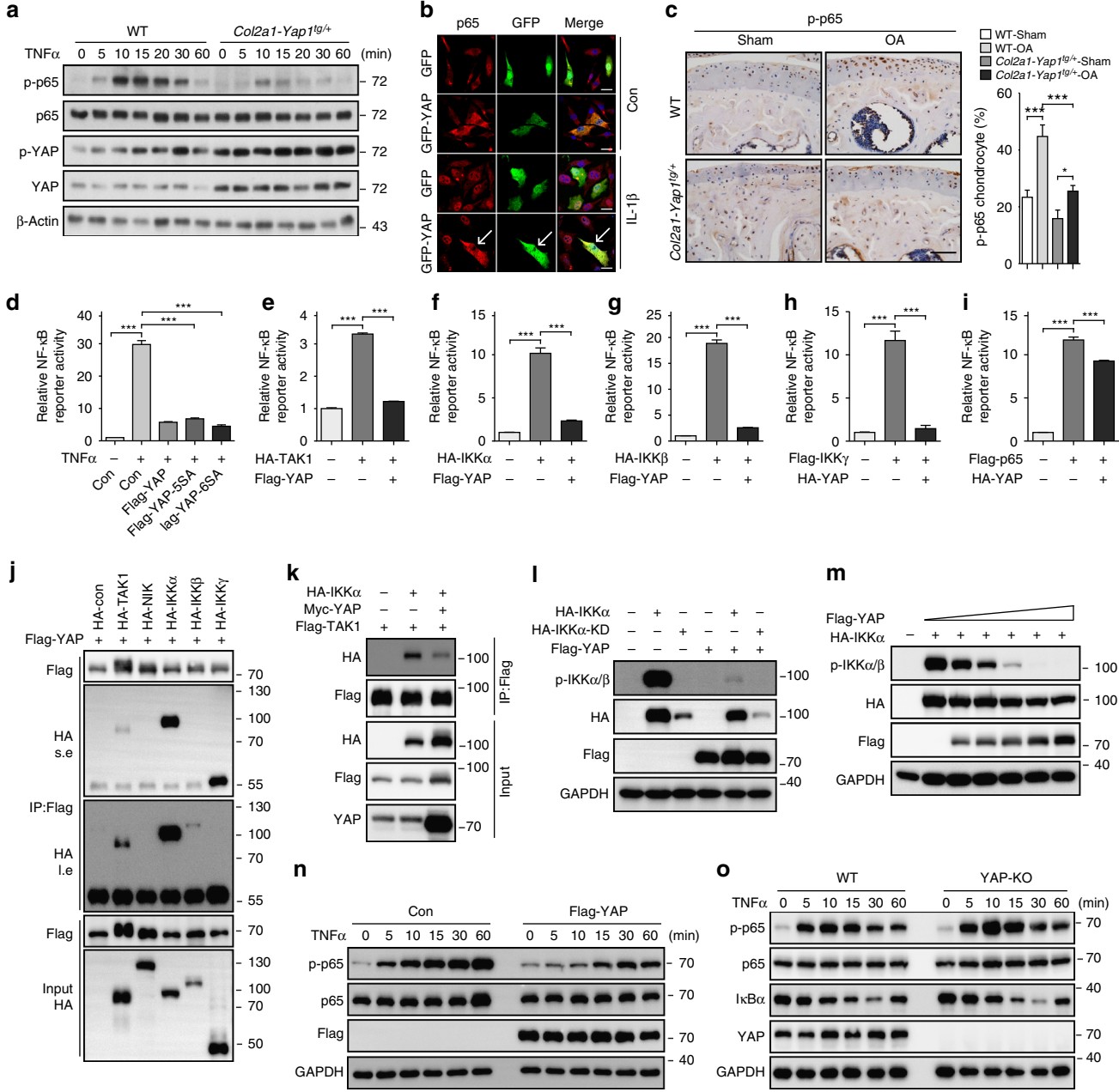

**Fig. 7** YAP attenuates NF-κB signaling by inhibiting IKKα/β activation. **a** Immunoblot analysis of primary chondrocytes with genotypes as shown after treatment with TNFα at 5 ng/ml. **b** Immunofluorescence analysis for p65 cellular localization in primary chondrocytes transfected with GFP-tagged YAP for 24 h and followed by treatment with IL-1β at 5 ng/ml for 1 h. Arrows indicate primary chondrocytes positively transfected with GFP-tagged YAP. **c** Immunohistochemistry of phospho-p65 expression in articular cartilage of mice 2 months after ACLT surgery with genotypes as shown. Scale bars, 50 μm. Statistical analysis of the percentage of p-p65 positive chondrocytes in articular cartilage of samples are shown in the right panel. **d** Luciferase assay of NF-κB luciferase reporter in HEK293A cells transfected with YAP, YAP-5SA, or YAP-6SA plasmid followed by treatment with TNFα for 6 h before collecting cell lysate. **e–i** Luciferase assay of NF-κB luciferase reporter 24 h after transfection of YAP with respective NF-κB component plasmids as indicated in HEK293T cells. **j** Immunoprecipitation assay to detect the association of YAP with indicated kinases related to NF-κB signaling in HEK293T cells. s.e. short time exposure of film. l.e. long time exposure of film. **k** Immunoprecipitation assay of IKKα and TAK1 with or without overexpression YAP in HEK293T cells. **l** Immunoblot analysis of the phosphorylation of IKKα with overexpression of YAP in HEK293T cells. **m** Immunoblot analysis of the phosphorylation of IKKα with expression of different dose of YAP in HEK293T cells. **n** Western blot analysis of lysate of HEK293A cells transfected with Flag-tagged YAP after treatment with TNFα as indicated time. **o** Immunoblot analysis of the phosphorylation of endogenous p65 after treatment with TNFα as indicated time in WT or YAP-KO HEK293T cells. All data are presented as mean ± SD. *p < 0.05, **p < 0.01, ***p < 0.001. One-way ANOVA followed by Tukey's test was performed. All experiments were repeated three times independently

Intriguingly, YAP overexpression abrogated the association of TAK1 with IKKα (Fig. 7k) and the phosphorylation of IKKα and IKKβ was significantly impeded by YAP or TAZ expression, respectively, in HEK293T cells (Fig. 7l, m, and Supplementary Figure 8h-k). All of these data suggest that YAP/TAZ antagonize NF-κB activity by preventing TAK1 substrate accessibility and subsequent activation of IKKα/β.

Activated IKKs complex induces phosphorylation and degradation of IκBα and subsequent activation of p65 and translocation of p50/p65 heterodimer into the nucleus. As β-TRCP complex is the ubiquitin ligase for ubiquitination of IκBα[27–29], we first examined whether YAP impeded the interaction of IκBα with β-TRCP. The ubiquitination of IκBα and the interaction between IκBα and β-TRCP were greatly reduced when YAP was overexpressed (Supplementary Figure 8l, m). Next, we examined p65 activation in the presence of YAP. Consistent with the reduced IKKα/β activation, the phosphorylation of p65 was attenuated in response to TNFα stimulation when YAP or TAZ was overexpressed in HEK293A cells, respectively (Fig. 7n and Supplementary Figure 8n). In addition, purified GST-YAP protein from *Escherichia coli* was sufficient to inhibit the phosphorylation of p65 by IKKα/β as revealed by the in vitro kinase assay (Supplementary Figure 8o, p). Together, our findings indicate that YAP inhibits TAK1 substrate accessibility, abrogates IKKα/β activation and suppresses IκBα degradation, and p65 activation and nuclear translocation, which results in attenuation of NF-κB signaling activity.

To further investigate the role of YAP in NF-κB signaling more specifically, we knocked down *Yap1* in primary chondrocytes isolated from newborn mice. TNFα treatment in *Yap1*-knockdown chondrocytes resulted in consistent results as in the *Yap1f/f; Col2a1-Cre* mutant chondrocytes with increased expression of Adamts4/5 and Mmp genes (Supplementary Figure 9a). Next, we generated YAP-KO HEK293T cells using CRIPSR/Cas9 system. There was an increased phosphorylation of p65 and IKKα/β in the YAP-KO HEK293T cells (Fig. 7o and Supplementary Figure 9b). In line with the results from YAP-knockdown chondrocytes, complete removal of YAP in the HEK293T cells resulted in a more sensitive response to TNFα stimulation as reflected by NF-κB luciferase reporter assay (Supplementary Figure 9c). These results indicate that YAP deficiency potentiates NF-κB signaling activity.

Collectively, our results suggest a reciprocal antagonism between Hippo-YAP/TAZ and NF-κB signaling in articular chondrocytes during the OA pathogenesis (Supplementary Figure 9d). Inflammatory cytokines promote YAP proteasomal degradation via TAK1-mediated YAP phosphorylation and ubiquitination. Reciprocally, YAP inhibits cartilage degradation through association with TAK1 and IKKs complex, which prevents the IKKα/β activation and subsequent NF-κB translocation into the nucleus.

## Discussion

This study shows that YAP mediates the function of Hippo pathway to control articular cartilage homeostasis during OA by antagonizing NF-κB signaling. In particular, genetic removal of both *Mst* kinases in chondrocytes displays the same phenotypes as YAP transgenic mice in protecting articular cartilage. This effect can also be achieved by LPA treatment, which activates YAP during osteoarthritic condition. By contrast, removal of *Yap1* in chondrocytes exaggerates cartilage degradation. Upon excessive mechanical stress, which is common risk factors for OA, inflammatory cytokines such as TNFα or IL-1β are secreted and trigger NF-κB signaling activation. Our study reveals that these inflammatory cytokines are able to activate the Hippo pathway. Since extracellular growth factors responsible for the activation of the Hippo pathway remain largely unclear, our study provides new insight in the control of

the Hippo pathway. Previous studies showed that mechanical stimuli such as cell–cell contact and high cell density activate Hippo pathway and inhibit YAP activity. These effects are mediated through junction proteins such as α-catenin[30,31], polarity proteins[32], GPCR signaling[21], or mechanical stress[33]. Our findings further unravel additional signaling cues that are able to activate Hippo pathway in regulating tissue homeostasis and provide insights into how Hippo-YAP/TAZ signaling coordinate cell growth and various environmental stimuli in physiological and pathological conditions.

Our current study shows that YAP associates with TAK1 to prevent IKKα/β activation, and thus inhibiting NF-κB signaling (Supplementary Figure 9d). It has been previously shown that YAP can be phosphorylated by LATS1/2 subsequently undergo proteasomal degradation mediated by β-TRCP complex after CK1-mediated phosphorylation[25]. Alternatively, YAP undergoes lysosomal degradation upon phosphorylation by IKKε at S431[18]. Here, we demonstrate that TAK1 is able to directly phosphorylate YAP at multiple sites independently of LATS and subsequently presents and interacts with the complex with β-TRCP for proteasomal degradation. Our finding reveals another mechanism on regulating the activity of YAP and delineates the reciprocal antagonistic relationship between Hippo and NF-κB signaling. In our model, YAP attenuates NF-κB signaling activation and results in reduced ECM degradation by inhibiting the induction of matrix-degrading enzymes. Indeed, a number of NF-κB inhibitors have been reported to play anti-inflammatory functions in OA. For instance, glucocorticoids are potent NF-κB inhibitors that induce the expression of IκBα and increase cytosolic retention of NF-κB[34,35]. In our study, YAP or its activator also inhibits TNFα-induced IκBα degradation and NF-κB nuclear localization. Thus, it is not surprising that YAP preserves cartilage integrity by modulating NF-κB signaling activity, which is a major pathway implicated in OA pathogenesis. In addition, it has been recently shown that cactus, a *Drosophila* IκBα homolog, is a direct target of Yorkie in regulating innate immunity system in fruit fly[36]. Interestingly, a recent study showed that YAP activates NF-κB activity through suppression of USP31, a negative regulator of NF-κB signaling, in soft tissue sarcomas[37]. These findings suggest that the crosstalk between Hippo pathway and NF-κB signaling are highly conserved in regulating inflammatory responses with distinct mechanism. Of note, our data also demonstrate that YAP/TAZ interfere with IKKα/β activation and IκBα stability in the mammalian system by inhibiting TAK1 function. Indeed, TAK1 inhibitor 5Z-7-O has been shown to have therapeutic values in inflammatory diseases[38]. As such, simultaneous targeting of Hippo and NF-κB pathway or more specifically YAP/TAK1 is a plausible strategy for better outcome of OA therapy.

Previous studies showed that Lubricin, a proteoglycan that is abundantly expressed in articular cartilage, which functions to protect cartilage integrity[39,40], also regulates the expression and localization of NF-κB during OA[41]. It seems that regulation of NF-κB pathway is a key mechanism in controlling articular cartilage homeostasis. Our findings that Hippo pathway or YAP modulates NF-κB signaling activity provide additional insights on how the NF-κB signaling activity could be fine-tuned or precisely controlled during OA pathogenesis. Recent studies demonstrate a direct regulation between Hippo-YAP/TAZ and innate antiviral pathways, which are involved kinases such as TBK1 and IKKε kinases[17,18]. Our study uncovered TAK1/IKKα/β kinases and YAP/TAZ complex as a hub linking inflammatory responses triggered by NF-κB signaling and chondrocyte proliferation and survival regulated by YAP/TAZ during OA pathogenesis. Interestingly, TAK1, TBK1, and IKKε directly interact with YAP/TAZ, and YAP/TAZ substantially suppress the activity of both TAK1 and TBK1. Our study also showed that TAK1 phosphorylates YAP/TAZ at multiple sites

independent of LATS1/2, although the functional significance of these sites was not investigated in the current study. Further work is required to identify and clarify the TAK1-mediated YAP/TAZ phosphorylation. Structural analysis on how YAP interacts with TAK1, TBK1, or IKKα/β will offer new insight into the molecular basis of inhibitory effect of YAP on these kinases.

During postnatal cartilage growth and osteoarthritic-induced cartilage degradation, we observed that YAP expression was gradually reduced. This expression pattern is highly similar to that of chromatin protein Hmgb2 (High mobility group box protein 2), a marker of chondroprogenitor cells in the articular cartilage that function to maintain cartilage integrity[42,43]. As YAP also regulates chondroprogenitor cell proliferation and is required for its maintenance[16], it is tempting to suggest that YAP can also be used as a marker for the integrity of articular cartilage. Furthermore, both YAP and Hmgb2 are transcriptional regulators and it is possible that they work co-operatively to regulate articular cartilage homeostasis. Interestingly, Hmgb2 has been identified as a potential transcriptional target of YAP/TEADs in a genome-wide ChIP assay[44]. Their specific mechanistic interactions warrant further investigation.

Our previous work demonstrated that YAP regulates multiple steps during chondrocyte differentiation and maturation and it is also implicated in bone repair[16]. In contrast to its inhibitory role in bone repair, our current study shows that YAP attenuates cartilage degradation during OA progression and preserves cartilage integrity. Indeed, YAP plays similar roles in inhibiting chondrocyte maturation during skeletal development, postnatal cartilage growth, and fracture healing. The impairment of cartilaginous callus formation during bone repair is a result of inhibition of chondrocyte maturation. However, during early stage of bone repair, cell proliferation of chondroprogenitor cells and early committed chondrocytes are increased with YAP expression. Similar results were obtained in ATDC5 cells[45], and C3H10T1/2 cells[46]. In addition, a recent study also showed that YAP expression is upregulated in the Gdf5-lineage cells in the synovium during cartilage repair[47]. The cell population contributing to cartilage repair is dependent on and regulated by YAP. All these findings are consistent with our current study that YAP plays positive roles in cartilage repair.

Our work here showed that inflammatory cytokines trigger YAP/TAZ degradation mediated by TAK1. However, both YAP and TAK1 have been implicated in response to mechanical stimuli. TAK1 is activated upon cartilage injury[48] and YAP activity is inhibited upon mechanical stress[33]. It seems that the trigger on TAK1/YAP action is multifactorial in OA development, in which both mechanical stress and inflammatory stimuli will result in similar actions on TAK/YAP. As recent views suggest that mechanical injury predisposes cartilage to OA development and it is believed to be the most critical etiologic factors[49,50], and that mechanical injury also triggers the secretion of inflammatory cytokines, our findings in the current study may also be contributed by excessive mechanical load. The involvement of TAK1/YAP actions stimulated by both mechanical and inflammatory stimuli raises the importance of this pathway to be a potential target for therapeutic strategy for OA treatment.

## Methods

**Human subjects.** Tibial plateau and femoral condyle specimens from human subjects with osteoarthritis undergoing total knee joint replacement surgery were collected with the approval by the Department of Health of Hong Kong and The Institutional Review Board of The Chinese University of Hong Kong. Full written consents were obtained before the operative procedure. The specimens were processed for histological examination and were categorized according to the International Cartilage Repair Society (ICRS) scoring system.

**Mice.** The Mst1[f/f][51], Mst2[f/f][51], Yap1[f/f][52], Col2a1-Cre[53], and Col2a1-Yap1[16] mouse lines have been described previously. All animal experiments were performed according to procedures approved by the Animal Experimentation Ethics Committee of the Chinese University of Hong Kong and Zhejiang University.

**Experimental OA animal models.** ACLT or DMM surgery was performed using 10-week-old male C57/B6 mice at the right knee joint[54,55]. Briefly, 12-week-old mice undergo the anterior cruciate ligament transection (ACLT) surgery via an incision on the medial para-patellar of the right knee joint capsule with longitudinal incision on the anterior cruciate ligament (ACL) and menisci resection. The destabilization of the medial meniscus DMM surgical instability models of osteoarthritis is similar to the ACLT surgery but without resection of menisci. A sham operation with the control mice of the same age was performed with a similar incision at the right joint capsule without anterior cruciate ligament incision and menisci resection. Animals were killed at 4 or 8 weeks after surgery. Dissected joints were processed for either histopathological or molecular analysis.

**Chemicals and reagents.** Verteporfin (VP) (SML05434), MG132 (C2211), 5Z-7-Oxozeaenol (O9890), and IL-1β (SRP6169 (human) and SRP8033(mouse)) were purchased from Sigma. SP600125 (T3109) was purchased from TargetMol. Lysophosphatidic acid (LPA) (sc-201053) was purchased from Santa Cruz. Cycloheximide (CHX) (2112 s) and TNFα (5178 (mouse) and 8902 (human)) was purchased from Cell Signaling. The Phos-tag TM Acrylamide AAL-107 was purchased from the NARD Institute. Kinase-dead TAK1 (TAK1-KD) construct was created by mutating lysine 63 to tryptophan. Kinase-dead IKKα (IKKα-KD) construct was created by mutating lysine 44 to alanine. Kinase-dead IKKβ (IKKβ-KD) construct was created by mutating lysine 44 to methionine.

**Cell culture and transfection.** HEK293A (ATCC), HEK293T (ATCC), and HeLa (ATCC) cells were cultured in DMEM medium with 10% FBS. All the cells were cultured at 37 °C in cell culture incubator with humidified environment in 5% $CO_2$. Lipofectamine 3000 (Invitrogen) or polythylenimine (PEI) (Polysciences) transfection reagents were used for plasmid transfection. All cell lines used were tested for mycoplasma contamination using the Mycoplasma Test Kit (Shanghai Yise Medical Technology Co. Ltd, PM008).

**CRISPR/Cas9-mediated genomic editing for knockout constructs.** Guide RNA sequences targeting TAK1, LATS1, LATS2, or YAP were cloned into the plasmids PX459 (Addgene #62988). Constructs were transfected into HEK293A or HEK293T cells by PEI transfection reagent. Twenty-four hours after transfection, cells were selected by puromycin (1.5 μg/ml) for 72 h. Single colonies were picked and identified by immunoblotting. Guide RNA, shRNA, and siRNA sequences are listed in Supplementary Table 1.

**Isolation of articular chondrocytes.** Primary chondrocytes were isolated from the P1–P3 newborn mouse according to the previous protocol[56]. Briefly, skin and soft tissues were removed from the hind limb of the pups. Tibial plateau, femoral heads, and femoral condyles were dissected and harvested under sterile conditions. The collected tissues were washed in PBS for 20 min at 37 °C. The cartilage tissue was digested using Collagenase type I (200 U/ml, Roche) for 20 min at 37 °C to remove the soft tissue. Cartilage pieces were retrieved and incubated with digestion buffer (Collagenase type I (200 U/ml) and Collagenase type D (0.5 mg/ml)) for 60 min at 37 °C. After digestion, cell suspension was subjected to centrifugation at 400×g for 5 min. Primary chondrocytes were seeded in culture plates. Only the first passage cells were used for experiments.

**RNA extraction and quantitative RT-PCR.** Total RNA from cultured cells was isolated using TRIzol Reagent (Invitrogen, USA) according to the manufacture's protocol. Two microgram total RNA was reversely transcribed using M-MLV Reverse Transcript Kit (Invitrogen, USA). Quantitative PCR was then performed using SYBR Green 2X PCR Master Mix (Applied Biosystems) on an Applied Biosystems 7900 system (Applied Biosystems, USA). Target gene threshold cycles (Ct values) were normalized to Gapdh as an endogenous control. The sequences of the primers are listed in Supplementary Table 1.

**Luciferase assay.** Primary chondrocytes or HEK293T cells were transfected with either pGL3-basic, NF-κB or Gal4/TEAD4-luciferase reporter plasmid together with pRL-TK vector (Promega, USA) as reference controls using Lipofectamine 3000 (Invitrogen, USA). Cells were subjected to luciferase activity measurement as described in Dual luciferase reporter assay kit (Promega, USA). NF-κB reporter plasmid was a gift from Dr. Zongping Xia (Zhejiang University). Gal4/TEAD4 luciferase reporter was a gift from Dr. Bin Zhao (Zhejiang University).

**SDS-PAGE, Phos-tag SDS-PAGE, and immunoblot analysis.** Gels for SDS-PAGE or Phos-tag SDS-PAGE were prepared according to the manufacturer's instructions (NARD Institute). Equal amount of protein was loaded into each well and separated by SDS-PAGE or Phos-tag SDS-PAGE. After blocking in 5% milk or

3% BSA, the membrane was incubated with primary antibody at 4 °C overnight. The membrane was washed and subjected to incubation with the HRP-conjugated secondary antibody for 2 h at room temperature. Antibody information is described in Supplementary Table 2. Uncropped blot images are provided in Supplementary Figure 10.

**Immunoprecipitation**. After transfection with indicated plasmid for 24 h, HEK293T cell lysates were collected on ice. Twenty microliter Anti-Flag M2 magnetic beads or Anti-Myc Agarose Affinity Gel antibody was added to the lysate, incubated at 4 °C for 8 h under gentle agitation. The beads were washed with washing buffer for five times. Finally, the beads were eluted with $2 \times$ SDS loading buffer. The eluted protein was analyzed by SDS/PAGE and followed by immunoblot analysis.

**Ubiquitination assay**. Primary articular chondrocytes or HEK293T cells were transfected with plasmids expressing Flag-YAP, Flag-IκBα, or HA-ubiquitin. Cells were pre-treated with MG132 at 10 μM for 2 h and then treated with TNFα at 5 ng/ml. Cell lysates were harvested and rocked with 20 μl Anti-Flag M2 magnetic beads at 4 °C for 8 h. The beads were washed five times. Bound proteins were boiled and analyzed by SDS/PAGE followed by immunoblot analysis. HA-ubiquitin plasmid was donated from Dr. CHEUNG Wing Tai (The Chinese University of Hong Kong). HA-K63-ubiquitin plasmid was a gift from Dr. Zongping Xia (Zhejiang University).

**In vitro kinase assay**. HA or Flag-tagged TAK1, IKKα/β, p65 and HA-YAP proteins were purified from HEK293T cells. GST-tagged YAP proteins were purified from E.coli by glutathione agarose slurry and eluted with glutathione. Purified proteins were washed with kinase washing buffer (40 mM Hepes and 200 mM NaCl, pH 7.5) for three times, and once with kinase assay buffer (30 mM Hepes, 50 mM KAC, and 5 mM MgCl₂, pH 7.5). Purified kinase and YAP proteins were mixed with ATP or ATP-γ-S (500 μM) in kinase assay buffer. After 30 min kinase reaction in 30 °C, EDTA (final concentration 20 mM, pH 8.0) was added to terminate the reaction at 30 °C for 5 min. Then, PNBM (final concentration 2.5 mM) was added to form a thiophosphate ester side chain at 25 °C for 40 min. Western blot was performed using anti-Thiophosphate ester antibody or phospho-specific antibody to analyze the kinase activity.

**Nano-liquid chromatography-tandem mass spectrometry analysis**. Nano-liquid chromatography-tandem mass spectrometry (nano-LC-MS/MS) analysis for protein identification and label-free quantification was performed by Phoenix National Proteomics Core service[57]. YAP proteins were trypsin digested at 37 °C for 16 h and desalted using C18 Zip-Tips (Millipore) according to the manufacturer's instructions. Next, the samples were resuspended in water and subjected to LC-MS/MS. The samples were separated using in-house made 12 cm length reverse phase columns (150 μm id) packed with Ultimate XB-C18 1.9 μm resin (Welch materials). The liquid chromatography was performed using an Easy nanoLC system (Thermo Fisher Scientific, USA). Phosphorylated peptides and site quantitation were carried out based on the extracted ion current chromatograms of the monoisotopic peaks. Tandem mass spectra corresponding to the putative phosphorylation-modified peptides and sites were verified by manual inspection of their fragmentation pattern.

**Immunofluorescence and imaging analysis**. Primary articular chondrocytes or HeLa cells were plated on coverslips. After treatment with TNFα or IL-1β at 5 ng/ml for 0, 15, 30, or 60 min, cells were then fixed with pre-cooled methanol on ice for 20 min. Cells were then washed with PBST (PBS, 0.1% Triton X-100), followed by blocking with 3% BSA at room temperature for 30 min. Cells were incubated with primary antibodies for YAP or p65, diluted in 3% BSA at 1:100 for 2 h at room temperature. After three washes with PBST, cells were incubated with secondary antibodies in 3% BSA for 2 h. Nuclei were stained with 0.5 μg/ml of DAPI (PBST) at room temperature for 10 min. Cells were mounted with ProLong® Gold Antifade (P36930, Thermo Fisher) and viewed with a Confocal system with inverted microscope (Olympus FV1000). Images were analyzed using the scientific software module of Imaris.

**Safranin O staining**. Safranin O staining was carried out according to the previous protocol (IHC World). The bone section slides were rehydrated to distilled water and stained with Weigert's iron hematoxylin for 10 min. After washing in running water, the slides were stained with 0.05% Fast Green solution. The slides were put into 1% acetic acid directly without washing and incubated for no more than 15 s. Slides were stained with 0.1% Safranin O solution for 5 min. After washing in running water for 1 min, the slides were dehydrated and cleared with 95% ethyl alcohol, absolute ethyl alcohol, and xylene. Slides were mounted using resinous mounting medium.

**Immunohistochemistry**. Immunohistochemistry was performed according to the manufacture's protocol of Histostain-Plus IHC Kit (Invitrogen, USA). Briefly, after deparaffinization and rehydration, slides were quenched in 3% H₂O₂ in methanol,

rinsed three times in PBS. The slides were subjected to antigen retrieval using Trypsin for 20 min at 37 °C. After washing 3 times in PBS, the slides were incubated with blocking reagent for 30 min. The slides were incubated with primary antibodies, biotinylated secondary antibodies, enzyme conjugated substrate and developed with diaminobenzidine (DAB) chromogen.

**Statistical analysis**. The comparisons between multiple groups, such as OARSI scores, were performed using multiple comparisons by one-way ANOVA followed by Turkey's test. For qRT-PCR data expressed as relative fold changes, Student's t-test and one-way ANOVA with Dunnett's test were used for pairwise comparisons and multi-group comparison, respectively. Results are represented as mean ± s.d. $p$ values < 0.05 were considered to be significant. Equal variances were assumed. All analyses were performed with GraphPad Prism software (Version 6.0). No statistical methods were used to predetermine sample size. The experiments were not randomized and the investigators were not blinded to allocation during experiments and outcome assessment.

## Data availability

The mass spectrometry proteomics data have been deposited to the ProteomeXchange Consortium via the PRIDE partner repository with the dataset identifier PXD011256[58]. All data supporting the conclusions are either provided in the manuscript or available from the authors upon request.

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

## Acknowledgements

We thank members of the Song's and Mak's groups for stimulating discussion. We thank Drs. Zongping Xia, Pinglong Xu, and Bin Zhao at Life Sciences Institute, Zhejiang University for the reagents and helpful discussion. This work was supported by the National Natural Science Foundation of China (grant number 31471368) and the Zhejiang Provincial Natural Science Foundation of China (grant number LR16C120001) to H.S., and the National Natural Science Foundation of China (grant number 81700002) and the International Postdoctoral Research Fellowship Program to Y.D., and the Seed Fund of the School of Biomedical Sciences, The Chinese University of Hong Kong (4620504) to K.M. H.S. is a scholar in the National 1000 Young Talents Program.

## Author contributions

Y.D., K.M., and H.S. designed the experiments. Y.D. performed most of the experiments; J.L. contributed to the in vitro kinase assay; W.L. and X.Z contributed to the DMM surgery, A.W. helped in breeding of Yap1^{f/f};Col2a1Cre mouse model; W.T., L.Q., and K. H. contributed to human specimen collection. H.S. and K.M. wrote the manuscript. All authors provided editorial comments.
