## [Peer Review File · Nature Communications]

Reviewers' comments:

Reviewer #1 (Remarks to the Author):

This is an outstanding manuscript shedding novel insights into the reciprocal relationship between TAK1 and YAP/TAZ. The authors present very solid convincing evidence for this novel interaction and are honest about the bits that are as yet unclear. My enthusiasm for the paper is partly due to the emerging concept of a reciprocal relationship between cartilage degradation and repair. The published data would agree that TAK1 is driving cartilage degradation in OA and YAP has recently been implicated in repair (Roelofs and de Bari, Nat Comm 2017). There is no mention of repair in the discussion or introduction which is remiss in view of their findings and the emerging realisation of this in human and murine disease.

Whilst TAK1 is certainly a key driver of disease, I do not share their view that degradation is necessarily NFkB driven. Indeed our own studies dissecting the pathways downstream of TAK1 found that IL1-mediated aggrecanase activity was due to JNK2 activation, not NFkB (Ismail et al, 2015 <http://doi.org/10.1002/art.39099>, Ismail et al, 2016 <http://doi.org/10.1002/art.39547>). Of course it is likely that some of the inducible metalloproteinases such as MMP13 are NFkB regulated at the mRNA level.

I also do not share their view that the evidence supports a role for inflammatory cytokines in OA pathogenesis. There are now several negative studies (published and unpublished) showing that IL1 and TNF are not driving OA pathogenesis in surgical models of OA (Clements et al, <http://doi.org/10.1002/art.11355>, Fukai et al, <http://doi.org/10.1002/art.33324> (hidden in discussion)). Moreover, there are several RCTs showing that targeting IL1 or TNF in patients is unsuccessful in knee and hand OA (Chevalier, <http://doi.org/10.1038/nrrheum.2013.44>). I think the authors are missing a trick by not stressing that mechanical injury of cartilage alone is sufficient to activate TAK1, induce down stream inflammatory signalling and control cartilage degradation (Ismail et al, 2017 <http://doi.org/10.1002/art.39965>, Vincent et al, <http://doi.org/10.1002/art.20369>). Mechanical load is essential for murine OA development after surgical joint destabilisation as well as induction of inflammatory genes (Burleigh et al, <http://doi.org/10.1002/art.34420>). The paper, in my opinion would be greatly improved by embracing these newer concepts in OA pathogenesis.

Reviewer #2 (Remarks to the Author):

This paper reports on the role the Hippo/Yap/Taz pathway in articular cartilage and osteoarthritis (OA). Results show that genetic removal of both Mst1/2 kinases in chondrocytes, or transgenic overexpression of YAP in mice preserves articular cartilage integrity in mouse models of OA. Loss of YAP in chondrocytes promotes cartilage disruption. Inflammatory cytokines, such as TNF α or IL-1 β , trigger YAP/TAZ degradation through TAK1-mediated phosphorylation. Furthermore, YAP directly interacts with TAK1 and attenuates NF- κ B signaling by inhibiting substrate accessibility

NOVELTY

The Hippo/Yap/Taz pathway is well characterized in regulating chondrocyte differentiation at multiple steps during endochondral ossification and bone repair. Its role in regulating inflammation is also known.

A large part of the result section addresses mechanisms that regulate Yap/Yaz (phosphorylation, degradation, relationship with TAK1). Most (if not all) of these events have been examined in other cell models. Authors need to highlight which of the findings are novel. Similar limitations concern the studies about Yap and attenuation of NFkB signaling.

DATA QUALITY AND INTERPRETATION

Overall, the experimental approaches are sound. The in vivo data are generated by using several different strains of mutant mice. The in vitro data are also solid, as they are generated by using several independent experimental approaches.

The data in figure 1 show reduction in Yap expression in mice between 1 -6 months. This is interpreted as indicating an ageing-related reduction. This interpretation is incorrect as it reflects postnatal cartilage growth and maturation related changes. To address ageing related changes mice from age 6 to 24 months should be analyzed. Data OA-related changes are conclusive.

Page 6 lines 2-3 the statement 'Altogether, our findings suggest that YAP plays an important role in the pathogenesis of OA.' Is not correct. The data just show a correlation between changes in Yap expression and OA.

The same laboratory previously reported that Yap regulates growth plate chondrocyte differentiation. The role of Hippo pathway in regulating articular chondrocyte differentiation should also be addressed (at least in the discussion) as this is also abnormal in OA. Two prior publications (PMID:28438716 and PMID:26025096) which also address the role of YAP in chondrocyte differentiation should also be discussed.

More detail needs to be provided how normal articular cartilage formation in the various strains of mutant mice was assessed as these were constitutive and not postnatally induced mutant mice.

Reciprocal antagonism between YAP/TAZ and NF- κ B signaling during osteoarthritic cartilage degradation

Yujie Deng^{1,2}, Jinqiu Lu¹, Wenling Li², Ailing Wu¹, Xu Zhang², Wenxue Tong³, Kiwai Kevin Ho³, Ling Qin³, Hai Song^{1,*}, Kinglun Kingston Mak^{2,*}

Our specific and detailed responses to the reviewers' comments are as follows:

Reviewer #1:

- This is an outstanding manuscript shedding novel insights into the reciprocal relationship between TAK1 and YAP/TAZ. The authors present very solid convincing evidence for this novel interaction and are honest about the bits that are as yet unclear. My enthusiasm for the paper is partly due to the emerging concept of a reciprocal relationship between cartilage degradation and repair. The published data would agree that TAK1 is driving cartilage degradation in OA and YAP has recently been implicated in repair (Roelofs and de Bari, Nat Comm 2017). There is no mention of repair in the discussion or introduction which is remiss in view of their findings and the emerging realisation of this in human and murine disease.

We thank the reviewer's suggestion and we have added the implication of YAP during bone and cartilage repair in the discussion on page 20, line 20 and added the suggested reference.

- Whilst TAK1 is certainly a key driver of disease, I do not share their view that degradation is necessarily NF κ B driven. Indeed our own studies dissecting the pathways downstream of TAK1 found that IL1-mediated aggrecanase activity was due to JNK2 activation, not NF κ B (Ismail et al, 2015 <http://doi.org/10.1002/art.39099>, Ismail et al, 2016 <http://doi.org/10.1002/art.39547>). Of course it is likely that some of the inducible metalloproteinases such as MMP13 are NF κ B regulated at the mRNA level.

To address the reviewer's concern, we treated primary chondrocytes with TNF α in combination with either TAK1 inhibitor 5Z-7-O or JNK inhibitor SP600125 (Supplementary Fig. 7d). We found that co-treatment of 5Z-7-O efficiently inhibited the stimulated effect of TNF α in inducing *Adamts* and *MMPs* expressions. By contrast, SP600125 treatment was not able to suppress their expressions except *Adamts5* and *Mmp3*. Similarly, 5Z-7-O treatment significantly inhibited TNF α - or IL-1 β - induced p65 phosphorylation, a key component of the NF- κ B signaling as well as JNK activation (Supplementary Fig. 7b, 7c). However, SP600125 treatment only inhibited JNK

activation (Supplementary Fig. 7b, 7c). More importantly, YAP overexpression inhibited both JNK and p65 phosphorylation upon TNF α stimulation (Fig.7a and Supplementary Fig. 7a). These data suggest that NF- κ B signaling is not the only downstream pathway to mediate the effects of TAK1/YAP, but may possibly also through the JNK pathway. Apart from the biochemical, cellular and histologic analyses of p65, we have also shown that the effects of YAP on NF- κ B signaling activity by luciferase reporter assay. We also demonstrated the association of YAP and its regulatory effects on key kinases of the NF- κ B signaling. Therefore, NF- κ B signaling is at least in part involved in the downstream actions of YAP.

- I also do not share their view that the evidence supports a role for inflammatory cytokines in OA pathogenesis. There are now several negative studies (published and unpublished) showing that IL1 and TNF are not driving OA pathogenesis in surgical models of OA (Clements et al, <http://doi.org/10.1002/art.11355>, Fukai et al, <http://doi.org/10.1002/art.33324> (hidden in discussion)). Moreover, there are several RCTs showing that targeting IL1 or TNF in patients is unsuccessful in knee and hand OA (Chevalier, <http://doi.org/10.1038/nrrheum.2013.44>).

We agree with the reviewer's point that TNF α or IL-1 β may not be the driving factors in OA pathogenesis, and more likely it is secondary to cartilage degradation. Mechanical injury triggers and induces cytokines expression. Here, we use TNF α or IL-1 β treatment for *in vitro* analyses in chondrocytes to mimic the conditions during OA pathogenesis and to study the mechanism of YAP degradation. We have toned down the effects of inflammatory cytokines in OA pathogenesis and discussed the role of mechanical injury in OA development in our revised manuscript.

- I think the authors are missing a trick by not stressing that mechanical injury of cartilage alone is sufficient to activate TAK1, induce down stream inflammatory signalling and control cartilage degradation (Ismail et al, 2017 <http://doi.org/10.1002/art.39965>, Vincent et al, <http://doi.org/10.1002/art.20369>). Mechanical load is essential for murine OA development after surgical joint destabilisation as well as induction of inflammatory genes (Burleigh et al, <http://doi.org/10.1002/art.34420>). The paper, in my opinion would be greatly improved by embracing these newer concepts in OA pathogenesis.

We thank the reviewer for the suggestions and we have added the importance of mechanical injury and activation of TAK1 in the discussion on page 21, line 11.

Reviewer #2:

- A large part of the result section addresses mechanisms that regulate Yap/Yaz (phosphorylation, degradation, relationship with TAK1). Most (if not all) of these events have been examine in other cell models. Authors need to highlight which of the findings

are novel. Similar limitations concern the studies about Yap and attenuation of NFκB signaling.

The novelty of our manuscript is the link of TAK1/YAP in OA development and pathogenesis. The function of YAP in OA progression has not been reported and our work demonstrates that YAP protects articular cartilage from degradation in OA. In terms of novel mechanistic insight, we showed that inflammatory cytokines is able to trigger YAP inactivation and degradation and this is mediated through TAK1. Our results showed that TAK1 directly phosphorylates YAP and such action promotes YAP degradation through β-TRCP complex. We also showed that YAP reciprocally inhibits the TAK-IKK signaling cascade. These findings shed new insights on the regulatory mechanism on the induction of matrix-degrading enzymes in the context of chondrocytes, which is critical for cartilage degradation during OA pathogenesis. Although the crosstalk between Hippo and NF-κB pathways has been implicated in several studies, our work revealed the reciprocal antagonistic relationship between YAP/TAZ and NF-κB signaling and explored the molecular mechanisms on how YAP/TAZ interacts with TNFα-induced NF-κB pathway. To our knowledge, this is the first study showing that YAP/TAZ activity can be regulated by TAK1 kinase and that YAP/TAZ suppresses TAK1 kinase activity. Our study identified a new regulator of YAP/TAZ proteins independent Hippo pathway.

- The data in figure 1 show reduction in Yap expression in mice between 1 -6 months. This is interpreted as indicating an ageing-related reduction. This interpretation is incorrect as it reflects postnatal cartilage growth and maturation related changes. To address ageing related changes mice from age 6 to 24 months should be analyzed. Data OA-related changes are conclusive.

We apologize for the mistake. We have changed the description and referred to postnatal cartilage growth and maturation in the revised manuscript on page 5, line 10 and 18.

- Page 6 lines 2-3 the statement ‘Altogether, our findings suggest that YAP plays an important role in the pathogenesis of OA.’ Is not correct. The data just show a correlation between changes in Yap expression and OA.

We apologize for the mistake. We have changed a more appropriate description in the revised manuscript on page 6, line 4.

- The same laboratory previously reported that Yap regulates growth plate chondrocyte differentiation. The role of Hippo pathway in regulating articular chondrocyte differentiation should also be addressed (at least in the discussion) as this is also abnormal in OA. Two prior publications (PMID:28438716 and PMID:26025096) which also address the role of YAP in chondrocyte differentiation should also be discussed.

We thank for the reviewer's suggestion and we have added the role of YAP in regulating chondrocyte differentiation in the discussion on page 20, line 20 and the suggested references have also been discussed on page 21, line 6.

- More detail needs to be provided how normal articular cartilage formation in the various strains of mutant mice was assessed as these were constitutive and not postnatally induced mutant mice.

We have included additional time points for Safranin O staining from 1 month old up to 4 months old to demonstrate the normal articular cartilage formation in various strains of the mutant mice in the revised manuscript as suggested (Supplementary Fig 1a, 1b; Supplementary Fig 2a, 2b and Supplementary Fig 3a, 3b)

REVIEWERS' COMMENTS:

Reviewer #1 (Remarks to the Author):

The authors have significantly improved the manuscript by being a little more circumspect about the role of inflammatory cytokines in OA (for which there is little direct evidence). The manuscript results contribute significantly to our understanding of OA pathogenesis.

Reviewer #2 (Remarks to the Author):

Authors provided satisfactory revisions to address this reviewer's comments.